# Dietary Copper Deficiency Leads to Changes in Gene Expression Indicating an Increased Demand for NADH in the Prefrontal Cortex of the Rat’s Brain

**DOI:** 10.3390/ijms23126706

**Published:** 2022-06-16

**Authors:** Monika Cendrowska-Pinkosz, Marta Ostrowska-Lesko, Katarzyna Ognik, Magdalena Krauze, Jerzy Juskiewicz, Anna Dabrowska, Jaroslaw Szponar, Slawomir Mandziuk

**Affiliations:** 1Department of Human Anatomy, Medical University of Lublin, 4 Jaczewski Street, 20-090 Lublin, Poland; monika.cendrowska-pinkosz@umlub.pl; 2Department of Toxicology, Medical University of Lublin, 8b Jaczewski Street, 20-090 Lublin, Poland; anna.dabrowska@umlub.pl; 3Department of Biochemistry and Toxicology, Faculty of Animal Sciences and Bioeconomy, University of Life Sciences in Lublin, 13 Akademicka Street, 20-950 Lublin, Poland; katarzyna.ognik@up.lublin.pl (K.O.); magdalena.krauze@up.lublin.pl (M.K.); 4Division of Food Science, Institute of Animal, Reproduction and Food Research, Polish Academy of Sciences, 10 Tuwim Street, 10-748 Olsztyn, Poland; j.juskiewicz@pan.olsztyn.pl; 5Toxicology Clinic, Faculty of Medicine, Medical University of Lublin, 100 Krasnik Avenue, 20-718 Lublin, Poland; jaroslaw.szponar@umlub.pl; 6Department of Pneumology, Oncology and Allergology, Medical University of Lublin, 8 Jaczewski Street, 20-090 Lublin, Poland; slawomir.mandziuk@umlub.pl

**Keywords:** copper deficiency, gene expression, metabolism

## Abstract

Copper is an essential element to brain cells as it is a cofactor and a structural component of various enzymes involved in energy metabolism pathways. Accumulating evidence points to the pivotal role of copper deficiency in neurodegeneration resulting from impaired copper homeostasis. Despite the indisputable role of copper in mitochondrial respiration, its homeostasis regulation in the brain tissue remains unclear. The assessment of changes in the expression of genes encoding key pathways of energy metabolism can greatly benefit further studies exploring copper’s role in neurodegeneration. Using a rat model, we investigate whether the replacement of the inorganic form of copper with metallic nanoparticles containing copper or complete deprivation of copper from the diet have an impact on the expression of genes involved in energy metabolism in the prefrontal cortex of the rats’ brain. Herein, we indicate that removing inorganic copper from the normal standard diet or the replacement with copper nanoparticles can lead to programmed energy metabolism changes. It can be recognized that some of these changes indicate an increased demand for NADH in the prefrontal cortex of the rat’s brain, probably as a result of adaptation effect.

## 1. Introduction

Copper (Cu) is a trace element found in almost all cells of the human body [1]. High concentration of Cu may be found in the brain, liver, and kidney [1]. Cu is involved in the regulation of many biochemical processes, including an iron metabolism and hemoglobin synthesis [2,3,4,5]; immune response [6,7,8]; proper functioning of the nervous system, in which it plays an important role in nerve myelination; synthesis of norepinephrine; and endorphin activity [9,10]. In addition, Cu is a catalytic cofactor of enzymes (Cu-dependent enzymes, also known as ‘cuproenzymes’), which are crucial for the redox balance that have implications in energy metabolism and antioxidant defense [11,12,13], for example cytochrome C oxidase; the terminal oxidase in most aerobic organisms that reduces molecular oxygen (O_2_) to water [10]; and superoxide dismutase (SOD1 and SOD3) [14]. Cu deficiency may also limit or affect the function of Cu-binding proteins [11].

Cu is highly toxic in excess, which results in an increase in the generation of free radicals [14,15]. There are known diseases characterized by both deficiency and overload of Cu in tissues: Menkes and Wilson’s diseases [15,16]. Cu deficiency in the body may lead to connective tissue diseases, muscle weakness, hypochromic anemia, and immune system dysfunction [16]. Moreover, there are reports on changes in the Cu homeostasis in the brain during neurodegenerative diseases, such as major depressive disorder (MDD), autism spectrum [17], Alzheimer’s disease [18,19,20], bipolar disorder, depression, ADHD, and paranoid schizophrenia [21].

Cu is an essential component of the mitochondrial complex IV (cytochrome C oxidase) [10,11]. In Johnson and Newman’s study (2003), a Cu-deficient diet in rats resulted in a decrease in cytochrome C oxidase, an increase in manganese superoxide dismutase (MnSOD), and glutathione peroxidase (GPX) in isolated cardiac mitochondria [22]. Chen et al. study (2002) demonstrated impaired cardiac mitochondrial respiration in rats with a Cu-deficient diet [23]. Famitafreshi and Karimian (2019) study on MDD model showed the decrease in Cu in the prefrontal cortex was accompanied by changes in glucose metabolism [24].

For these reasons, this study attempts to assess whether the replacement of the inorganic form of Cu with nanoparticles containing Cu or complete deprivation of Cu from the diet have an impact on the expression of genes involved in energy metabolism in the prefrontal cortex of the brain.

## 2. Results

In order to establish the impact of the absence and type of Cu in the diet on energy metabolism of the brain prefrontal cortex, we measured changes in gene expression using rats treated with diet of different microelement mixture (see Section 4 for details).

The results show that the Cu deprivation has significant impact on changes in the expression of tested genes. Considering genes involved in glycolysis, we observed increased expression of *Hif1a* and *Hk2* genes and slightly decreased expression of *Myc* and glucose transporter *Slc2a1*, which is one of its targets (Figure 1a).

The experimental conditions did not affect *Ldha* expression. Further analysis demonstrated changes in the expression of genes encoding mitochondrial proteins: decreased expression of *Pc* (encoding enzyme catalyzing of pyruvate conversion to oxaloacetate); increased expression of *Mpc1* (transport of pyruvate to mitochondria), *Pdk4*, *Mdh1*, *Ppara*, *Me*, *Cpt2*, *Got2* (conversion of oxaloacetate to aspartate), *Gls*; and higher expression of genes encoding antioxidant enzymes *Sod2* and *Cat* (catalase) (Figure 1b–d). A graphical presentation of the impact of Cu deprivation from the diet on selected metabolic pathways is illustrated in Figure 2. For details see Appendix A.

The results show that the replacement of the mineral Cu with a metallic form of Cu in the diet does not affect the expression of the *Hif1a* gene and leads to a reduction in the expression of the *Myc* gene. However, there was no significant changes in the activity of genes under the *Myc* control: *Slc2a1*, *Hk2*, nor *Ldha* (Figure 3a).

A very similar pattern of changes in the expression of some genes encoding mitochondrial proteins was found in Cu_MNP_ group: decreased *Pc* expression, and increased mRNA expression of *Mpc1*, *Pdk4*, *Mdh1*, *Idh2*, *Ppara*, *Gls*, *Got2*, *Sod2*, and *Cat* genes (Figure 3b–d). However, a different direction of changes in expression was observed for *Cpt2* expression (involved in fatty acid oxidation)—the replacement of CuCO_3_ by Cu nanoparticle caused a decrease in *Cpt2* mRNA expression. A graphical presentation of the impact of the replacement of the mineral Cu with the metallic form of Cu in the diet on selected metabolic pathways is illustrated in Figure 4. Details can be found in Appendix A.

The elimination of CuCO_3_ from a standard diet leads to the activation of genes responsible for obtaining NAPH directly or indirectly. This is evidenced by the activation of the genes involved in glucose phosphorylation (*Hk2*), the mitochondrial transport of pyruvate (*Mpc1*), the mitochondrial transformation of oxalate to aspartate (*Got2*), the mitochondrial transformation of glutamine into glutamate (*Gls*), and an increase in *Cpt2* expression, which enables the fatty acid β-oxidation—an essential source of NADH in mitochondria.

## 3. Discussion

Worldwide, over 10 million people suffer from neurological disorders annually [25]. This number is expected to rise in the future. The increase in oxidative stress associated with aging; low activity of enzymes, such as catalase or glutathione peroxidase; as well as the presence of unsaturated fatty acids in cell membranes make the human brain susceptible to neurodegeneration [26,27]. Moreover, studies have shown that a nutrient-poor diet can lead to disturbances in the central or peripheral nervous system. Neurodegenerative disorders mainly develop as a result of protein misfolding. For example, misfolding of the τau and amyloid-β proteins lead to the progression of Alzheimer’s disease [27].

The American Nutraceutical Association defines nutraceuticals as a food or product with health-promoting properties [28]. Evidence shows that nutraceuticals have emerged as a promising strategy in the treatment of several chronic diseases, including neurological disorders [28]. Nutraceuticals can enhance the therapeutic effect of some drugs when used in combination by enhancing several pathways, thus providing unique neurobiological effects [29].

The above research focuses on changes in the expression of genes involved in selected pathways of energy transformations during exposure to Cu deficiency or the conversion of Cu from inorganic to metallic form, which may greatly benefit further studies exploring Cu’s role in neurodegeneration. We also investigated changes in the mRNA expression level for molecules responsible for the redox balance since interference with the physiological balance of metabolic transformations is associated with the risk of changes/disturbances in the redox balance, as well as the role of Cu for the activity of many enzymes. Additionally, the expression of well-known genes controlling metabolic pathways—*Hif1α* and *Myc*—was demonstrated. To the best of our knowledge, such studies have not been conducted so far.

### 3.1. Changes Caused by the Copper Deprivation from the Diet

The most important observation from our studies in the prefrontal cortex of Cu-deficient rats are changes in the expression of genes associated with mitochondrial metabolism, indicating an increased need/production of NADH.

*Slc2a1* is the gene encoding the glucose transporter 1 (GLUT-1), located in the plasma membrane. After glucose is transported inside the cell, glucose is phosphorylated by HK2 and in this form cannot leave the cell. Cu deprivation from the diet did not alter the expression of the gene for the glucose transporter (*Slc2a1*) located in the membrane of the cells of the prefrontal cortex. Nevertheless, gene expression for the enzyme responsible for glucose phosphorylation, *Hk2*, increased. However, the conducted studies do not explain whether the phosphorylation of the cellular glucose pool increases with an unchanged rate of its intracellular transport, or whether the increased phosphorylation of glucose is dependent on the increase in glucose-intracellular transport. It seems that the lack of changes in *Slc2a1* expression does not exclude the possibility of enhanced intracellular glucose transport. It can be suspected that the increase in intracellular transport, to some extent, may be due to the increased transport efficiency of the constitutive GLUT-1 molecules without the need to increase the synthesis of this transporter, as is the case in the Warburg Effect [30]. In addition, other GLUT isoforms may also be involved in the intracellular transport of glucose [31]. Thus, based on the evaluation of *Slc2a1* and *Hk2* expression, it may be concluded that these observations do not contradict each other. Therefore, an increase in glucose phosphorylation may be possible due to the increased efficiency of intracellular-glucose transport without changes in GLUT-1 expression.

If we assume that an increase in *Hk2* expression is needed to increase glucose phosphorylation, it may indicate an increased need to activate glycolysis (oxidation to pyruvate) and/or the pentose phosphate pathway, which is the source of pentoses and the main cellular source of the NADPH reducing factor. Depending on the needs, the cell increases the activity of the pentose phosphate cycle when the demand for NADPH (used for the synthesis of FFA and in maintaining antioxidant activity) increases and/or the demand for pentoses necessary for the synthesis of nucleotides, e.g., RNA, increases. On the other hand, the activation of glycolysis may result in the increased regeneration of NADH from NAD^+^ (a reaction catalyzed by glyceraldehyde-3-phosphate dehydrogenase) and an increase in pyruvate synthesis and intensification of its multidirectional transformations [32].

Pyruvate crosses the outer mitochondrial membrane through passive diffusion and is transported through the inner mitochondrial membrane into the matrix by Mitochondrial Pyruvate Carrier 1 (MPC1) [33,34]. Our research showed an increase in the expression of the *Mpc1* gene. If the increase in *Mpc1* mRNA expression is accompanied by the increased transport of pyruvate to the matrix, it can be used in direct synthesis to acetyl-CoA and oxaloacetate. Acetyl-CoA combines with oxaloacetate to form citrate, which is burned in the Krebs cycle or, after being transported to the cytoplasm, is re-converted to oxaloacetate and acetyl-CoA. The acetyl-CoA formed in the cytoplasm is the basic building block for the synthesis of fatty acids (acetyl-CoA cannot be directly transported from the mitochondria to the cytoplasm). Oxaloacetate itself in the cytoplasm can also be used in the gluconeogenesis pathway. However, there was no increase in the activity of genes responsible for mitochondrial pyruvate transformations, both to acetyl-CoA (*Pdk4*) and oxaloacetate (*Pc*), as well as the Krebs cycle enzyme (*Idh2*), in which citrate is reduced (resulting from the synthesis of acetyl-CoA and oxaloacetate). The enzyme encoded by *Pdk4* decreases conversion pyruvate to acetyl-CoA by phosphorylation of pyruvate dehydrogenase complex. Thus, it seems that there is no change in the regulation of pyruvate to acetyl-CoA conversion at the mRNA level, which does not mean that constitutive enzymes cannot increase catalytic activity depending on the energy/metabolic state in the cell without the need for additional enzyme synthesis. Further, a significant reduction in the expression of the *Pc* gene encoding Pyruvate Carboxylase (PC), responsible for the conversion of mitochondrial pyruvate to oxaloacetate, was found. It is probably associated with a decrease in the activity of this enzyme—a similar conclusion as to the relationship: gene expression/enzyme activity would be invalid if there was not a decrease but an increase in gene expression. Therefore, a conclusion can be made that a decrease in PC activity limits the production of mitochondrial oxaloacetate, and thus three metabolic pathways: 1. the Krebs cycle in the matrix; 2. the synthesis of lipids in the cytoplasm; 3. gluconeogenesis, which takes place in the cytoplasm but begins in the mitochondria, similar to the Krebs cycle, from the conversion of pyruvate to oxaloacetate. In this context, the increase in expression for the mitochondrial pyruvate transporter *Mpc1* remains elusive. However, with regard to the glycolytic pathway, an increase in the expression of the *Got2* gene in the matrix catalyzing the conversion of oxaloacetate to aspartate was found—it is a part of the malate-aspartate shuttle carrying oxaloacetate from the mitochondria to the cytoplasm. This is puzzling because there is minimal gluconeogenesis in the brain (mainly in the liver, the renal cortex, and in the muscle) [35]. Studies have also demonstrated evidence that gluconeogenesis exists in brain astrocytes, but no convincing data have yet been found in neurons [36]. Therefore, activation of the pathway at the GOT2 level may be associated with an increase in the demand for gluconeogenesis intermediators but used for purposes other than glucose synthesis, for example, for the synthesis of glycerol (glyceroneogenesis in adipose tissue) [37,38] and in the case of the brain, synthesis of neurotransmitters, and neuroglial pathway [39].

So, does the withdrawal of CuCO_3_ from the diet affect lipid changes in the prefrontal cortex? The removal of Cu from the diet leads to an increase in signals for both anabolic regulation and transport to the mitochondrial fatty acids, which may also indicate an increase in catabolic changes. Our research showed an increase in the expression of the gene encoding the transcription factor PPAR-α as well as *Cpt2* and *Me1*. It is generally known that activation of PPAR-α promotes uptake, utilization, and catabolism of fatty acids by upregulation of the genes involved in fatty acid transport, fatty acid binding and activation, and mitochondrial fatty acid β-oxidation [40,41]. These processes are predominantly localized in the liver [42], but in the brain, PPAR-α mainly takes part in regulation of genes coding proteins that are involved in glutamate homeostasis and cholinergic/dopaminergic signaling [43]. However, the effect of PPARα on lipid metabolism, e.g., peroxisomal proliferation, fatty acyl-CoA-oxidation, PUFA biosynthesis, HMG-CoA synthesis, and ketogenesis is also observed in the brain tissue [44].

Our research also showed an increased expression of *Me1* gene, which encodes the NADP-dependent malic enzyme catalyzing the malate to pyruvate transformation (NADPH for fatty acid biosynthesis is generated in this reaction) [45] and an increase in *Cpt2* expression allowing transport across the inner mitochondrial membrane [46] into the matrix, and thus β fatty acid oxidation. Thus, we observed the activation of lipid metabolic pathways with the simultaneous lack of activation of the mitochondrial conversion of pyruvate to oxaloacetate, used in the Krebs cycle, lipogenesis, or gluconeogenesis/glyceroneogenesis for the synthesis of glycerol and triglycerides. The level of expression of the *Gls* gene of the gluconeogenesis pathway, primarily expressed in the brain, showed an increased expression. *Gls* catalyzes the conversion of glutamine to glutamate, which is further converted to α-ketoglutarate, a compound of the Krebs cycle [47].

Summarizing, the assessment of the expression of genes related to metabolic transformations in the mitochondria indicates an increased demand for NADH, probably resulting from a decrease in energy production as a result of a decrease in cytochrome oxidase activity, in which Cu is a direct catalytic factor. This is evidenced by the increase in the expression of genes encoding the transport of pyruvate to the matrix, which, after transformation, becomes part of the Krebs cycle (the first NADH generator), and an increase in mRNA for *Gls1* and *Got2* that controls the pathway of glutamine conversion to oxaloacetate, which is included in the Krebs cycle, and an increase in *Cpt2* expression, which enables β oxidation of FFA (the second very important source of NADH in mitochondria). In addition, *Got2* regenerates NADH in mitochondria [48]. Moreover, the increase in *Hk2* expression is consistent with this thesis, as it may be an expression of an increased need to activate glycolysis in order to obtain pyruvate. In view of the current state of knowledge, it is difficult to explain the mechanism of these changes caused by a lack of Cu in the diet. Despite the increase in the demand for pyruvate, contrary to expectations, there is no increase in the signals regulating its use or even blocking the pathway of its use (decrease in *Pc* expression). It can only be assumed that the observed changes are an expression of pathophysiology disorders caused by a decrease in the activity of the enzymes of redox pathways because Cu is a key element in the catalytic activity of many of them [11,12]. In turn, among the changes of pyruvate taking place in the cytoplasm, the conversion to lactate is the most often described. Moreover, here, there was no increase in the expression of the *Ldha* gene, encoding the lactate dehydrogenase isoenzyme, composed solely of the “H” subunits (LDHA) responsible for the conversion of pyruvate to lactate [49]. Any changes in the rate of NADH consumption by the mitochondrial electron transport chain may alter the redox balance through ROS production [50].

NADPH is also used as the most important reducing agent in maintaining the redox balance in the face of excess ROS. It regenerates glutathione to a reduced (active) form that allows the antioxidant defense enzymes to be kept in a defensive readiness. In our research, in addition to the increased demand for NADH, we found an increase in the expression of genes encoding the malic enzyme that produces NADPH to maintain redox balance in mitochondria. This may indicate an increased need for NADPH not only for the synthesis of FFA but also/mainly to maintain the redox balance. Among the two tested genes encoding antioxidant enzymes, both SOD and catalase (CAT) increased gene expression. It cannot be ruled out that this is an adaptation of the cell to the growth of oxidants. In order to confirm this and to distinguish whether it is physiological, pathological, or oxidative stress compensation, in-depth research should be carried out.

It seems that the observed changes are not coordinated with each other (they escaped from cellular regulation), and are, therefore, accidental, which would rather indicate multidirectional disturbances in cellular regulation. However, contradictory changes in the expression of genes encoding glycolysis-regulating *c-myc* and *Hif-1a* transcription factors may explain some of these changes. In our study, *Hif1a* expression increased and *c-myc* expression decreased. HIF-1a factor induces the expression of the genes of proteins activating glycolysis. They include glucose transporters GLUT1, glycolytic enzymes (including hexokinase isoforms-HK, isoform M2 of pyruvate kinase), isoform of lactate dehydrogenase (LDHA) involved in lactate synthesis, and the lactate transporter MCT4. Indeed, in our study, *Hk2* expression was elevated. A very important role of HIF-1a is to reduce the production of acetyl-CoA from pyruvate produced in glycolysis. This increases the supply of pyruvate converted to lactate. The decrease in observed *Pc* expression may also increase the supply of pyruvate for lactate synthesis. However, no increase in *Ldha* expression was found. On the other hand, it is known that *c-myc* is a transcription factor that directly activates the genes of the GLUT1 transporter and glycolytic enzymes, such as HK2, phosprofructokinase-1, and enolase-1 [51]. Furthermore, *c-myc* is an activator of *Ldha* expression [51]. It can be concluded that the observed increase in *Hk2* expression and the lack of changes in *Ldha* may be the result of a simultaneous increase in *Hif1a* expression and a decrease in *c-myc* expression.

### 3.2. Changes Caused by Replacing the Mineral Copper with a Metallic Form in the Diet

The decrease in *Myc* expression did not result in a change in the regulation of the marked glycolytic genes (*Slc2a1*, *Hk2* and *Ldha*), which indicates a lack of negative genetic regulation of key molecules in the *Myc* glycolysis pathway. However, the transcription factor Myc (cMyc) regulates a number of other important processes for cell–cell proliferation, the target of growth, or apoptosis. It is worth emphasizing that neurons are post-mitotic cells where proliferation does not normally take place. However, the changes in diet from salt to metallic nanoparticles Cu induced during the experiment may create new, unusual conditions. In addition, in the sample collected for mRNA determination, there are also cells from other tissues, including: glia, cells of the immune system, or endothelium. Therefore, the measured mRNA expression is the resultant expression of mRNA from the cells of all these tissues. Such preliminary studies may indicate the need for further studies on glial, neuron, and endothelial cell cultures.

The analysis of changes in gene expression in the Cu_MNP_ vs. Cu_salt_ group encoding mitochondrial proteins indicates the possibility of reducing the activity of the Krebs cycle. The potential for the transport of pyruvate to the mitochondria increases, but the conversion of pyruvate to oxaloacetate decreases. On the other hand, the probability of converting the amount of oxaloacetate that is formed into aspartate, which is transported to the cytoplasm, is increased, and thus this amount of oxaloacetate is not directed to the Krebs cycle. In addition, the likelihood of fatty acid oxidation (decrease in *Cpt2* expression) decreases, which reduces the production of substrates for the Krebs cycle. In addition, an increase in the expression of genes encoding the antioxidant defense enzymes-CAT and SOD2 was observed, which may indicate redox imbalance towards the predominance of oxidants when we replace the standard CuCO_3_ diet with metallic Cu.

### 3.3. Limitations of the Study

The limitation of this study is sample size. According to the Three Rs principle (reduce, reuse, recycle; Directive 2010/63/EU), it was possible to use a maximum of 10 individuals per group. However, this number was sufficient to perform statistical calculations.

## 4. Materials and Methods

### 4.1. Nanoparticles Characterization

Copper nanoparticles (metallic form of copper, Cu_MNP_) were obtained from Sky Spring Nanomaterials, Inc. (Houston, TX, USA). They were 40–60 nm in size (nanopowder, 12 m^2^/g), with purity of 99.9%, spherical morphology, 0.19 g/cm^3^ bulk density, and 8.9 g/cm^3^ true density [28]. The zeta potential of Cu_MNP_ (30.3 mV) was determined previously by the Ognik et al. (2020) study [52].

### 4.2. Animal Breeding

All animal care and experimental protocols complied with the current laws governing animal experimentation in the Republic of Poland and by an ethical committee according to the European Convention for the Protection of Vertebrate Animals used for Experimental and other Scientific Purposes, Directive 2010/63/EU for animal experiments [53] approved by the Local Ethics Committee for Animal Experiments in Olsztyn, Poland (Permit Number: 68/2017).

Thirty, male albino Wistar rats (Han IGS Rat [Crl:WI(Han)]) aged 7 weeks with average body weight of 159 ± 2.65 g were randomly divided into 3 groups (*n* = 10). Rats were housed randomly and individually in stainless steel cages under a stable temperature (21–22 °C), relative humidity 50 ± 10%, a 12-h light-dark cycle, and a ventilation rate of 15 air changes per hour.

### 4.3. Dietary Treatment

For four weeks, the rats had free access to tap water and semi purified diets, which were prepared and then stored at 4 °C in hermetic containers until the end of the experiment (see Table 1).

The diets used in the experiment were modifications of the casein diet for laboratory rodents, recommended by the American Institute of Nutrition. The detailed composition of the mineral mixtures used in all experimental groups is presented in Table 2. The copper content of the diet samples was determined by inductively coupled plasma emission spectrometry (ICP-OES, Varian Inc., Palo Alto, CA, USA).

### 4.4. Experimental Design

To evaluate the effect of different sources of Cu in the diet, the animals were divided into three groups: Cu_salt_ (control group), rats fed with a standard diet as the mineral mixture containing 6.5 mg Cu/kg (CuCO_3_) for four weeks; Cu_D_ (experimental group), rats fed with a standard diet as the mineral mixture deprived Cu for four weeks; CU_MNP_ (experimental group), rats fed a standard diet as the mineral mixture, however Cu (6.5 mg/kg) was added to a diet not in mineral mixture, but as an emulsion along with dietary rapeseed oil. The detailed composition of the mineral mixtures used in all experimental groups has been provided in Table 3.

### 4.5. Material Collection

All physiological measurements were made separately for each animal. Rats were anesthetized by intraperitoneal injection of ketamine + xylazine (100 mg/kg + 10 mg/kg of body weight) according to recommendations for anesthesia and euthanasia of experimental animals. The rats were euthanized by cervical dislocation. Then the brain was removed and dissected. The prefrontal cortex samples were taken for further molecular analysis [52].

### 4.6. The Quantitative Real-Time PCR Analysis (qPCR)

#### 4.6.1. RNA Isolation

Total RNA was isolated using the TRIzol Reagent (Invitrogen, Waltham, MA, USA) and Chomczynski and Sacchi method [54]. Briefly, 15 mg tissue sections from the brain prefrontal cortex was homogenized in the presence of TRIzol Reagent, after which chloroform was added (POCH, Gliwice, Poland). The tube was centrifuged to separate the phases. Then, RNA was precipitated from the aqueous phase with isopropanol (POCH, Gliwice, Poland). The precipitated RNA was rinsed with 75% ethanol (POCH, Gliwice, Poland), then centrifuged and dissolved in RNase-free water (EURx, Gdansk, Poland). RNA concentration and purity were measured using the MaestroNano micro-volume spectrophotometer (Maestrogen, Hsinchu, Taiwan). High purity RNA (A260/280 ratio ranged between 1.8 and 2.0) was used for further cDNA synthesis.

#### 4.6.2. cDNA Synthesis

To obtain cDNA, RNA concentrations were normalized and the reverse transcription was performed using the NG dART RT-PCR kit (EURx, Gdansk, Poland) and random hexamers according to the manufacturer’s instructions. The reaction was carried out using a thermocycler (Mastercycler gradient, Eppendorf, Hamburg, Germany) and the following thermal profile: 10 min 25 °C, 50 min 50 °C, 5 min 80 °C. Each reaction was performed in triplicates. The obtained cDNA was stored at −20 °C. The sequences of used primers can be found in Appendix A.

#### 4.6.3. Real-Time PCR

The relative expression of studied genes was performed by real-time PCR method using 7500 fast real-time PCR system (AppliedBiosystems, Foster City, CA, USA). The procedures were carried out in accordance with the manufacturer’s protocols with minor modifications. The sample reaction mix consisted of PowerUp SYBR Green Master Mix (Applied Biosystems, Foster, CA, USA), RNase-free water and cDNA. Each reaction was carried out in three technical replications. The reaction profile was as follows: 95 °C for 2 min, 45 cycles: 95 °C for 15 s, 57 °C for 15 s and 72 °C for 1 min; melt curve 0.4 °C/s up to 97 °C. The quality of the samples was analyzed based on the amplification values, Tm and Ct to remove any deviations before calculating ΔΔCt and to determine the fold change in mRNA levels. Data on changes in gene expression at the mRNA level were presented as mean RQ ± SD.

### 4.7. Statistical Analysis

Data was analyzed by STATISTICA v.10 application (StatSoft, Cracow, Poland) and GraphPad Prism v.5.0 (GraphPad Software, San Diego, CA, USA). The results are presented as mean ± standard deviation. The normality of the distribution was analyzed using the Shapiro–Wilk test. The comparison between study groups were analyzed with a Student’s *t*-test, *p* ≤ 0.05 was considered to indicate a statistically significant difference.

## 5. Conclusions

Taken together, withdrawal the Cu from the diet or replacing inorganic Cu with metallic one may lead to the energy metabolism dysregulation in the brain prefrontal cortex. The changes in gene expression in rats fed with a Cu-free diet indicate the mobilization of energy metabolism pathways for increased NADH demand.

## Figures and Tables

**Figure 1 ijms-23-06706-f001:**
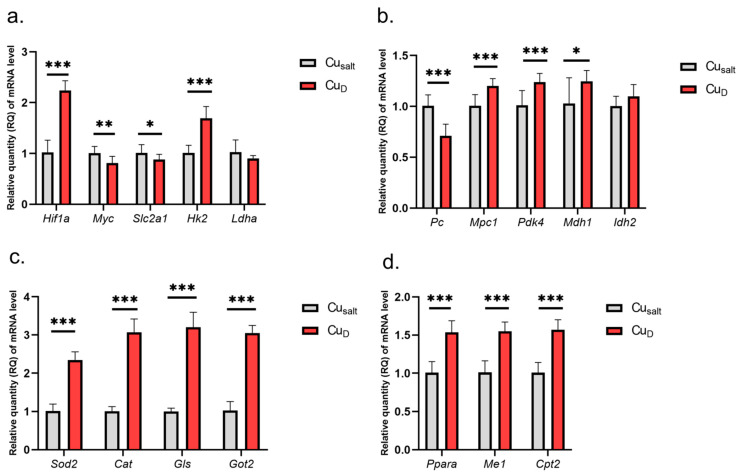
(**a**–**d**) Results of mRNA expression in a group of rats with copper deprivation. All values are reported as mean ± standard deviation of relative quantity of mRNA level (*n* = 10). Significance: * *p* ≤ 0.05, ** *p* ≤ 0.01, *** *p* ≤ 0.001 vs. control group (Student’s *t*-test). Cu_salt_, control group, treated with mineral mix; Cu_D_, group treated with mineral mix deprived copper.

**Figure 2 ijms-23-06706-f002:**
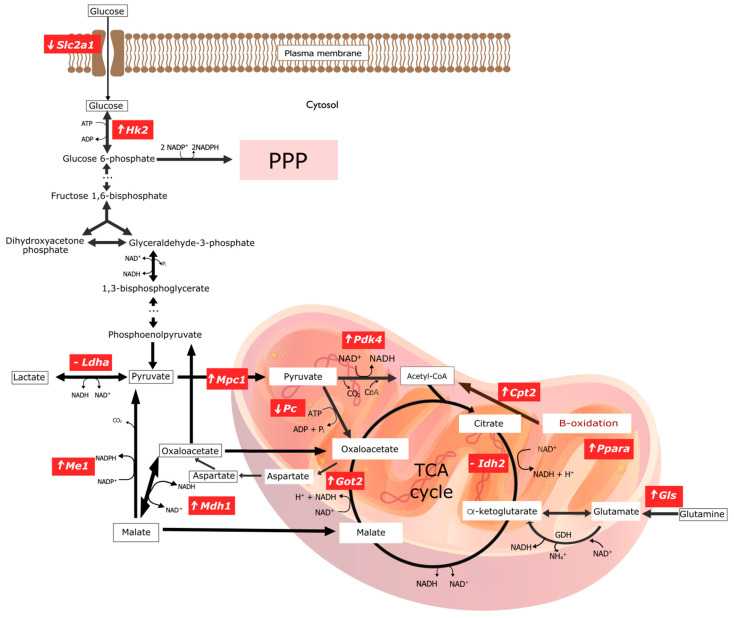
Graphical representation of changes in gene expression encoding key enzymes involved in selected pathways of energy transformations in experimental group of rats with copper deficiency. ‘⬆’ indicates overexpression of the gene encoding the given enzyme; ‘⬇’ indicates a downexpression of the gene encoding the given enzyme; ‘-’ means there are no significant statistical changes. The graphic was made in the open-source vector graphics editor InkScape. ADP, Adenosine diphosphate; ATP, Adenosine triphosphate; CO_2_, Carbon dioxide; CoA, Coenzyme A; *Cpt2*, Carnitine Palmitoyltransferase 2; GDH, Glutamate dehydrogenase; *Gls1*, Kidney-type glutaminase, mitochondrial; *Got2*, Glutamic-Oxaloacetic Transaminase 2; *Hk2*, Hexokinase 2; *Idh2*, Isocitrate Dehydrogenase (NADP^+^) 1; *Ldha*, Lactate Dehydrogenase A; *Mdh1*, Malate Dehydrogenase 1; *Me1*, Malic Enzyme 1; *Mpc1*, Mitochondrial Pyruvate Carrier 1; NAD^+^, Nicotinamide adenine dinucleotide; NADH, Nicotinamide adenine dinucleotide, reduced form; NH4^+^, Amonium; *Pc*, Pyruvate Carboxylase; *Pdk4*, Pyruvate Dehydrogenase Kinase 4; Pi, Pyruvic acid; *Ppara*, Peroxisome Proliferator Activated Receptor Alpha; PPP, Pentose Phosphate Pathway; *Slc2a1*, Solute Carrier Family 2 Member 1; TCA cycle, The tricarboxylic acid cycle.

**Figure 3 ijms-23-06706-f003:**
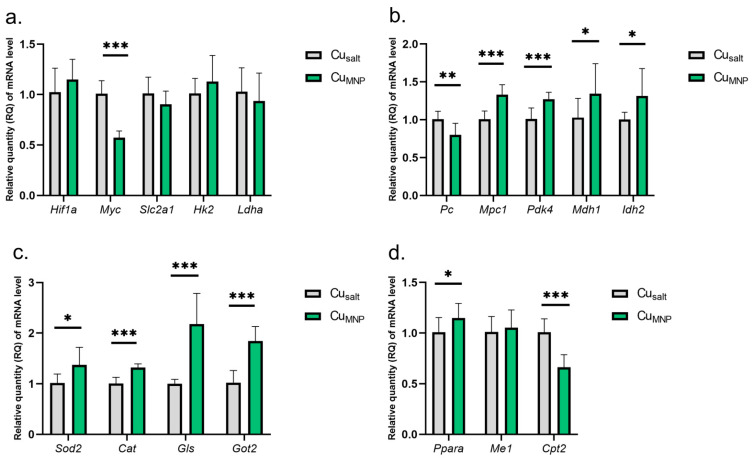
(**a**–**d**) Results of mRNA expression in a group of rats treated with the mineral copper with a metallic nanoparticle form in the diet. All values are reported as mean ± standard deviation of relative quantity of mRNA level (*n* = 10). Significance: * *p* ≤ 0.05, ** *p* ≤ 0.01, *** *p* ≤ 0.001 vs. control group (Student’s *t*-test). Cu_salt_, control group, treated with mineral mix; Cu_MNP_, group treated with mineral mix and 6.5 mg metallic Cu/kg from nanoparticles preparation as an emulsion along with dietary rapeseed oil.

**Figure 4 ijms-23-06706-f004:**
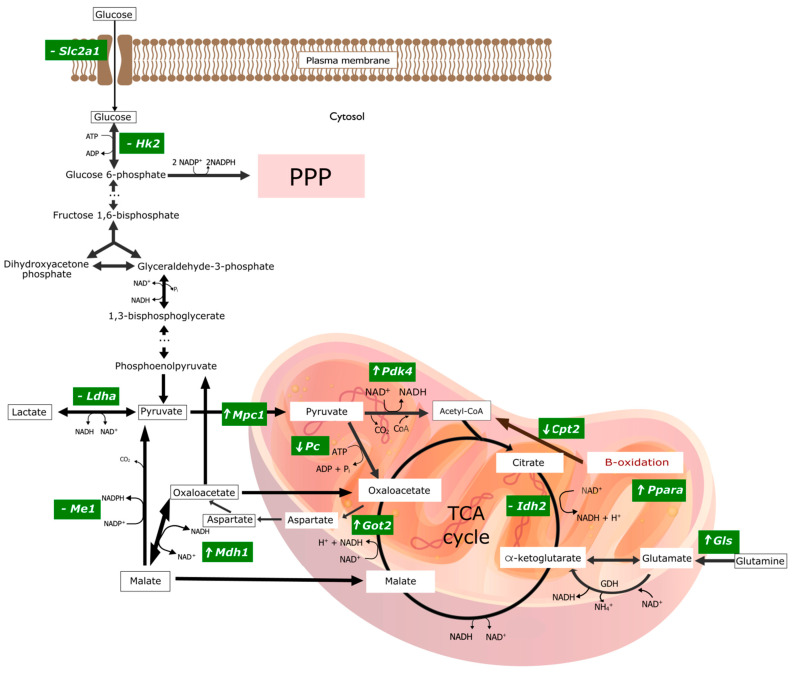
Graphical representation of changes in gene expression encoding key enzymes involved in selected pathways of energy transformations in experimental group of rats treated with the mineral copper with a metallic nanoparticle form in the diet. ‘⬆’ indicates overexpression of the gene encoding the given enzyme; ‘⬇’ indicates a downexpression of the gene encoding the given enzyme; ‘-’ means there are no significant statistical changes. The graphic was made in the open-source vector graphics editor InkScape. ADP, Adenosine diphosphate; ATP, Adenosine triphosphate; CO_2_, Carbon dioxide; CoA, Coenzyme A; *Cpt2*, Carnitine Palmitoyltransferase 2; GDH, Glutamate dehydrogenase; *Gls1*, Kidney-type glutaminase, mitochondrial; *Got2*, Glutamic-Oxaloacetic Transaminase 2; *Hk2*, Hexokinase 2; *Idh2*, Isocitrate Dehydrogenase (NADP^+^) 1; *Ldha*, Lactate Dehydrogenase A; *Mdh1*, Malate Dehydrogenase 1; *Me1*, Malic Enzyme 1; *Mpc1*, Mitochondrial Pyruvate Carrier 1; NAD^+^, Nicotinamide adenine dinucleotide; NADH, Nicotinamide adenine dinucleotide, reduced form; NH4^+^, Amonium; *Pc*, Pyruvate Carboxylase; *Pdk4*, Pyruvate Dehydrogenase Kinase 4; Pi, Pyruvic acid; *Ppara*, Peroxisome Proliferator Activated Receptor Alpha; PPP, Pentose Phosphate Pathway; *Slc2a1*, Solute Carrier Family 2 Member 1; TCA cycle, The tricarboxylic acid cycle.

**Table 1 ijms-23-06706-t001:** Composition of basal diet fed to rats, data presented as percent of content (%).

Ingredient	Content
Unchangeable ingredients	
Casein ^a^	14.8
DL-methionine	0.2
Cellulose ^b^	8.0
Choline chloride	0.2
Rapeseed oil	8.0
Cholesterol	0.3
Vitamin mix ^c^	1.0
Maize starch ^d^	64.0
Changeable ingredient	
Mineral mix ^e^	3.5
Calculated content	
Crude protein	13.5

^a^ Casein preparation: crude protein 89.7%, crude fat 0.3%, ash 2.0%, and water 8.0%. ^b^ α-Cellulose (SIGMA, Poznan, Poland), main source of dietary fiber. ^c^ AIN-93G-VM, g/kg mix: 3.0 nicotinic acid, 1.6 Ca pantothenate, 0.7 pyridoxine-HCl, 0.6 thiamin-HCl, 0.6 riboflavin, 0.2 folic acid, 0.02 biotin, 2.5 vitamin B-12 (cyanocobalamin, 0.1% in mannitol), 15.0 vitamin E (all-rac-α-tocopheryl acetate, 500 IU/g), 0.8 vitamin A (all-trans-retinyl palmitate, 500,000 IU/g), 0.25 vitamin D-3 (cholecalciferol, 400,000 IU/g), 0.075 vitamin K-1 (phylloquinone), 974.655 powdered sucrose. ^d^ Maize starch preparation: crude protein 0.6%, crude fat 0.9%, ash 0.2%, total dietary fiber 0%, and water 8.8%. ^e^ Changeable dietary ingredient in relation to Cu source (standard source CuCO_3_ and experimental: Cu deprivation and Cu nanoparticles preparation).

**Table 2 ijms-23-06706-t002:** Composition of mineral mixtures used in experimental diets, g/kg of mixture.

Ingredient	MX with Standard Cu Dosage ^a^	MX Deprived of Cu ^b^
Calcium carbonate anhydrous CaCO_3_	357.00	357.00
Potassium phosphate monobasic K_2_HPO_4_	196.00	196.00
Potassium citrate C_6_H_5_K_3_O_7_	70.78	70.78
Sodium chloride NaCl	74.00	74.00
Potassium sulphate K_2_SO_4_	46.60	46.60
Magnesium oxide MgO	24.00	24.00
Microelements mixture	18.00	18.00
Starch	213.62	213.62
Microelements mixture, g/100 g		
Ferric citrate (16.7% Fe)	31.00	31.00
Zinc carbonate ZnCO_3_ (56%Zn)	4.50	4.50
Manganous carbonate MnCO_3_ (44.4% Mn)	23.40	23.40
Copper carbonate CuCO_3_ (55.5% Cu)	1.85	0.00
Potassium iodate KJ	0.04	0.04
Citric acid C_6_H_8_O_7_	39.21	40.70

^a^ given to Cu_salt_ group (4 weeks of feeding); ^b^ given to Cu_D_, Cu_MNP_ groups (4 weeks of feeding); Cu_MNP_ group was provided with appropriate amount of Cu from Cu nanoparticles preparation as an emulsion along with dietary rapeseed oil; MX—mineral mix.

**Table 3 ijms-23-06706-t003:** The experimental scheme of applied copper.

Symbol of Group	Type of Group	Administered Diet
Cu_salt_	Control (*n* = 10)	Four-week-diet with mineral mix, containing 6.5 mg Cu/kg; Cu from CuCO_3_
Cu_D_	Experimental (*n* = 10)	Four-week-diet with mineral mix, deprived Cu
Cu_MNP_	Experimental (*n* = 10)	Four-week-diet with mineral mix, 6.5 mg Cu/kg; Cu from nanoparticles preparation

D, deficiency; *n* = 10, number of rats used in particular feeding period; MNP, metallic nanoparticle; S, salt. Provided Cu dosage was calculated taking into account CuCO_3_ in mineral mixture or Cu from Cu nanoparticles preparation).

## Data Availability

All relevant data are within the paper.

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
