# Peer review of "Dietary Copper Deficiency Leads to Changes in Gene Expression Indicating an Increased Demand for NADH in the Prefrontal Cortex of the Rat’s Brain"

_ijms, 2022, doi:10.3390/ijms23126706_

Round 1

Reviewer 1 Report

Title: Dietary copper deficiency leads to changes in gene expression indicating an increased demand for NADH in the prefrontal cortex of the rat's brain.”

-         Authors should include more numerical data in the results section of the abstract in order to improve its readability. Please update.

-         The English of the paper should be revised due to typos.

-         The role of nutraceuticals in cardiovascular health should be better described and specified. The authors can consider the paper from Scicchitano P et al. Journal of Functional Foods 2014;6:11-32.

Reviewer 2 Report

Introduction

The last paragraph should be rephrased to distinguish clearly between the hypothesis of the authors and the final objective of the work.

Procedures

2.1. Ten animals per group in this type of experiments is small. Please add a short passage in Discussion to balance this matter.

2.3. Please indicate in brief the method by which copper content in diets was measured.

Table 4. Please complete this table with all the details of the PCR (no need for any mention of details in the text) and please move to supplementary table.

2.7. In a supplementary table, please provide evidence for normality of data.

Results

Please in the figures, do not use white bars but bars with another colour.

Table 5: too detailed; please move to supplements and in Results replace with a coloured figure.

Discussion

Sub-section 4.1. please think if this needs to be subdivided to 4.1.1. and 4.1.2. for better flow of ideas.

Some recent (after 2019) references are missing.

Finally, I am not sure if figures are allowed in Discussion, please check with editorial office.

Overall. Corrections as indicated and re-evaluation.

Round 2

Reviewer 2 Report

The authors have improved the manuscript and have dealt with all queries and issues raised.
Before final acceptance, please look into the manuscript once again and correct some linguistic slips, which are scattered throughout the manuscript. Please use shorter sentences to give better flow of ideas to text and please correct various typos.
After that, the revised manus can be accepted.